# Different Characteristics and Clinical Outcomes between Early-Onset and Late-Onset Asthma: A Prospective Cohort Study

**DOI:** 10.3390/jcm11247309

**Published:** 2022-12-09

**Authors:** Bing-Chen Wu, Chiung-Hsin Chang, Yun-Chen Tsai, Tin-Yu Lin, Po-Jui Chang, Chun-Yu Lo, Shu-Min Lin

**Affiliations:** 1Department of Thoracic Medicine, Chang Gung Memorial Hospital, Taipei 333, Taiwan; 2School of Medicine, Chang Gung University, Taipei 333, Taiwan

**Keywords:** asthma, atopy, exacerbation

## Abstract

Late-onset asthma (LOA) differs from early-onset asthma (EOA) in terms of prognosis and the treatment response because it has a much worse prognosis and a poorer response to standard asthma treatment. This study sought to investigate the characteristics and clinical outcomes of asthma patients with phenotypes distinguished by age at onset and atopy status. We prospectively recruited patients with asthma who were registered in a pay-for-performance program operated by Taiwan’s National Health Insurance Administration (NHIA). These patients received regular outpatient treatment for at least 1 year at every outpatient clinic visit since 2019. Baseline characteristics and clinical outcomes were compared between patients with LOA (≥40 years) and those with EOA (<40 years). Of the consecutive 101 patients with asthma, 21 patients (20.7%) had EOA and 80 (79.3%) had LOA. In the 12-month period, patients with EOA had higher declines in forced expiratory volume in one second (FEV_1_; −2.1 ± 8.4 vs. 6.8 ± 13.1, % of predicted value, *p* = 0.037) and forced vital capacity (FVC; −4.6 ± 12.0 vs. 6.1 ± 13.6, % of predicted value, *p* = 0.023) than patients with LOA. Patients with nonatopic EOA had a significantly higher exacerbation rate at 12 months than patients with nonatopic LOA (50% vs. 11.8%, *p* = 0.012). Identification of different phenotypes of asthma is important in clinical practice because treatment responses may differ.

## 1. Introduction

Asthma is a heterogeneous disease with multiple phenotypes depending on the severity of symptoms, extent of airflow limitation, level of asthma control, frequency of exacerbations, type of underlying airway inflammation (eosinophilic or non-eosinophilic), and age at asthma onset [1]. The onset of asthma is often assumed to occur in childhood or early adulthood, but 30–40% of adults with asthma experience their initial symptoms after the age of 40 years. In addition, up to 52% of asthmatic patients have their “first asthma attack” after the age of 40 years [2,3]. Late-onset asthma (LOA) has a worse prognosis and a poorer response to standard asthma treatment compared with early-onset asthma (EOA) [4]. EOA and LOA are the most common phenotypes of asthma [5], and recent cluster-based studies have also identified age at onset as a key factor in distinguishing asthma phenotypes [6]. Atopic asthma represents the most common form of asthma in children [7], and it is often characterized by eosinophils and T helper 2 (Th2) cell-driven infiltration of the bronchial mucosa, circulating specific immunoglobulin E (IgE) antibodies, positive skin tests for common aeroallergens, and airway hyperresponsiveness [8]. Nonatopic asthmatics are significantly older than atopic asthmatics [8]. The etiologies of these phenotypes, including airway inflammation, natural progression, and responsiveness to treatment, are believed to differ [6]. According to a previous study, patients with nonatopic LOA are at a higher risk of developing persistent airflow limitation than those with atopy [9]. In contrast to patients with atopic EOA, patients with nonatopic EOA respond poorly to inhaled corticosteroids [10]. Identifying and evaluating these phenotypes will provide a comprehensive understanding of the causes and progression of asthma in adults, and this may facilitate the development of more targeted preventive strategies and treatment alternatives.

The clinical outcomes and response to treatment in asthma patients with different phenotypes of asthma and atopic status have been investigated. However, most of the studies are cross-sectional or population-based, focusing on the characteristics of different phenotypes in asthmatics. Therefore, collecting real-world data through longitudinal assessments of these phenotypes may provide a comprehensive understanding of clinical outcomes, such as the frequency of exacerbations and pulmonary function decline. However, the reliability of the results is influenced mainly by whether treatment management and patient compliance are standardized. The pay-for-performance program has been implemented by the National Health Insurance Administration (NHIA) in Taiwan since 2001 to strengthen the management and health education of asthma patients [11]. Furthermore, the NHIA urges certified physicians and case managers to provide in-person instructions on asthma control, asthma care planning, and proper inhaler use [12], and the outcomes are regularly monitored by the NHIA. Because of this, we could follow up patients throughout the study period and ensure the consistency of their treatment plans. This study sought to investigate the characteristics and clinical outcomes of asthma patients with phenotypes distinguished by age on onset and atopy status.

## 2. Materials and Methods

### 2.1. Study Design and Patient Recruitment

This was a prospective cohort study of asthma patients participating in a pay-for-performance program, who were recruited consecutively at Chang Gung Memorial Hospital, Linkou Branch. Patients were diagnosed with asthma by a pulmonologist on the basis of the concordance of respiratory symptoms, pulmonary function, and responsiveness to asthma treatment. After receiving their confirmed asthma diagnosis from a physician, they were followed up every 3 months for at least 1 year. The pay-for-performance program is implemented by the NHIA in Taiwan to strengthen the management and health education of asthma patients. For sample recruitment, patients who were registered in the pay-for-performance program and had a major diagnosis code (ICD-10 code J45) present at least twice within 90 days were consecutively enrolled into our study after providing their written informed consent. Patients with incomplete pulmonary function data, noncompliance with regular clinic visits, and coexisting chronic obstructive pulmonary disease (COPD) were excluded. The definition of adequate compliance was patients with adequate medication compliance by self-report of more than 80% use of prescribed medications and regular clinical visits. The pay-for-performance program was implemented in qualified medical centers in Taiwan by NHIA. This study recruited patients who were diagnosed as having asthma and received at least 2 clinic visits within 90 days in our hospital. Enrollment in the program was voluntary for patients, and they signed the patient agreement consent. The study protocol was approved by the Institutional Review Board of Chang Gung Memorial Hospital (No.201900211B0).

### 2.2. Patient Data

In this study, the following data were collected from patients on the basis of questionnaires or medical records: age at asthma diagnosis by a physician, history of pediatric dyspnea, frequency of bronchitis, gender, family history of asthma, smoking status, exacerbations in the previous year, and pulmonary function and T2 inflammation biomarkers, such as eosinophil count, eosinophil cation protein (ECP), total immunoglobulin E (IgE) level, and specific IgE (ImmunoCAP, Phadia, Uppsala, Sweden). Patients with any positive specific IgE to allergens (>0.35 kUA/L) were considered atopic [12]. We also recorded the number of acute asthma exacerbations, including hospitalization, emergency room visits, and systemic corticosteroid bursts, using medical records and patients’ self-reported data. The level of asthma control was evaluated using the asthma control test (ACT, a 5-question questionnaire, with each question scored from 1 to 5. The lower summed scores for all questions indicate poorer asthma control). Patients with LOA were defined as those who were ≥40 years old at the time of asthma onset without a history of pediatric dyspnea or bronchitis. Otherwise, they were classified as having EOA [1].

### 2.3. Outcome Measurement

A total of 101 patients were divided into 2 groups on the basis of the age at asthma onset. In each group, patients were further classified as having atopic and nonatopic asthma. Then, the clinical outcomes of each group at 1 year were compared. We performed spirometry in accordance with the American Thoracic Society (ATS) and European Respiratory Society (ERS) guidelines [13], recording forced expiratory volume in 1 s (FEV_1_) (% predicted), forced vital capacity (FVC), and FEV_1_/FVC. We also determined the number of asthma exacerbations (defined as a decline in FEV_1_ to <60% of the personal best, requiring an OCS burst and an unscheduled doctor/emergency room visit or hospitalization) and asthma-related hospitalizations during the 12-month period prior to recruitment and during the 12-month observation period.

### 2.4. Statistical Analysis

Unless otherwise indicated, all data are expressed as mean ± standard deviation or percentages. The Student’s *t*-test was used to compare the means of normally distributed continuous data between the 2 groups; otherwise, the Mann–Whitney test was used. The chi-squared test or Fisher’s exact test was used to examine the correlations between categorical variables. Statistical analyses were performed using SPSS software, version 26 (IBM Corporation, Armonk, NY, USA). The statistical significance level was set at a *p*-value of <0.05.

## 3. Results

### 3.1. Characteristics of Patients with EOA Compared with Patients with LOA

Patients in the pay-for-performance program were screened from April 2019 to December 2021. A total of 145 adult patients with asthma were enrolled. We excluded 16 patients because of missing regular asthma clinic visits, 17 patients because of missing spirometer measurements, and 11 patients because they had coexisting COPD. The remaining 101 asthma patients were eligible for analysis. Figure 1 provides the study flowchart. Among these asthmatics, 21 patients (20.7%) had EOA. The characteristics of patients with EOA and LOA are shown in Table 1. The average onset age of patients with EOA and LOA was 23.4 and 61.8 years, respectively. Patients with EOA were younger, had a longer duration of asthma, had a higher rate of family history of asthma, and had more comorbidities with allergic rhinitis or rhinosinusitis with or without polyps compared with patients with LOA. No difference was observed between the two groups in terms of gender, smoking habits, weight status, exacerbations in the last year, medications, or ACT score. The pre-bronchodilator FVC and FEV_1_ were higher in patients with EOA than in those with LOA. Regarding allergic status, the proportion of atopy was higher in the EOA group than in the LOA group.

The characteristics of patients with atopic and non-atopic status are shown in Table 1. Atopic patients were younger, had a lower BMI, had a longer duration of asthma, had a lower rate of male gender, and had more comorbidities with allergic rhinitis or obstructive sleep apnea compared with non-atopic patients. No difference was observed between the two groups in terms of smoking habits, family history of asthma, exacerbations in the last year, medications, or ACT score. The pre-bronchodilator FVC was higher in atopic patients than in non-atopic patients. Regarding allergic status, the IgE levels and rate of fungus sensitization were higher in the atopic group than in the non-atopic group.

### 3.2. Clinical Outcomes after 12-Month Treatment

Table 2 shows the pulmonary function at 12 months for patients with EOA and LOA. The mean difference in the percentage of predicted FVC and FEV_1_ from baseline declined in the EOA group, whereas it increased in the LOA group. Patients with EOA had greater declines in FEV_1_ (−2.1 ± 8.4 vs. 6.8 ± 13.1, % of the predicted value, *p* = 0.037) and FVC (−4.6 ± 12.0 vs. 6.1 ± 13.6, % of the predicted value, *p* = 0.023) than patients with LOA over a 12-month period (Figure 2). Both groups had similar baseline exacerbation rates. No difference was noted between the groups in clinical outcomes at 12 months. In the total asthma cohort (Figure 3), the incidence of total acute exacerbation events (28.7% vs. 12.9%, *p* = 0.009), emergency room visits (12.9% vs. 1.0%, *p* = 0.001), and hospitalizations (11.9% vs. 0%, *p* = 0.0003) decreased after 12 months of treatment. In the subgroups of atopic asthma patients, the incidence of AE decreased in both the EOA (20% (3/15) vs. 6.7% (1/15)) and LOA (29.4% (10/34) vs. 11.8% (3/34)) groups after 12-month treatment (Figure 4). In the subgroups of nonatopic asthma patients, the incidence of AE decreased in the LOA (32.4% (14/46) vs. 10.9% (5/46)) group after 12-month treatment (Figure 4). However, the incidence of AE increased in the EOA (33.3% (2/6) vs. 50% (3/6)) group after 12-month treatment. In the nonatopic subgroups, patients with EOA had a significantly higher exacerbation rate at 12 months than those with LOA (50% vs. 11.8%, *p* = 0.012). In contrast, the pulmonary function and acute exacerbation at 12 months were similar in the atopic and non-atopic groups (Table 2).

To elucidate the impact of age on clinical outcomes of patients with EOA and LOA, patients were stratified as age < and ≥ 65 years old. Among patients <65 years old (*n* = 43), there were 15 EOA patients and 28 LOA patients. The characteristics of patients with EOA and LOA are listed in Appendix A. In the subgroup of patients ≥65 years old, 6 patients were EOA and 52 patients were LOA. The baseline characteristics of patients ≥65 years old are shown in Appendix A. In patients <65 years old, EOA patients had greater FEV_1_ decline than LOA patients over a 12-month period (−5.3 ± 8.0 vs. 10.2 ± 12.3, % of the predicted value, *p* = 0.008) and FVC (−8.0 ± 12.7 vs. 8.4 ± 11.9, % of the predicted value, *p* = 0.008) (Appendix A; Figure 2B). In patients ≥65 years old, the changes of pulmonary function at 12 months were similar in EOA and LOA groups (Appendix A; Figure 2C).

### 3.3. The Treatments for Asthma in Each Group

Table 3 shows the treatment for patients with atopic and non-atopic status. In atopic patients, there was no difference in treatment for asthma in EOA and LOA groups. In non-atopic patients, there were more EOA patients who received OCS and biologics treatment than LOA patients.

## 4. Discussion

Our study results revealed that patients with EOA were younger, had a longer asthma duration, and had increased rates of allergic rhinitis, rhinosinusitis with or without polyps, and atopy than patients with LOA [14]. After 12 months of treatment, the rates of acute exacerbation events, including hospitalizations, emergency room visits, and systemic corticosteroid bursts, decreased in the total asthma cohort. The EOA group showed a greater decline in pulmonary function than the LOA group. Considering the atopic and asthma onset features, all three subgroups had a decreased incidence of AE, except that patients with nonatopic EOA had an increased incidence of exacerbations after 12 months of treatment.

This study found that after 12-month treatment, patients with EOA had an increased pulmonary function decline compared with patients with LOA. However, most previous studies have reported that patients with LOA exhibit a rapid decline in pulmonary function [9,15]. The possible reasons for the conflicting results regarding pulmonary function decline in patients with EOA may be due to the duration of asthma. In the present study, patients with EOA had a longer duration of asthma than patients with LOA. Long-lasting chronic airway inflammation may adversely affect pulmonary function [16,17]. In addition, the prevalence of airway remodeling appears to be higher in patients with EOA than in patients with LOA. The prominent presence of airway remodeling may inversely affect pulmonary function [18].

This study revealed a significant decrease in AE events after the 12-month asthma treatment in the total study cohort. Patients with EOA and LOA had similar responses to 12-month treatment in terms of the frequency of exacerbations. This may be due to high compliance and regular outpatient clinic follow-up visits. Our study results also showed that patients with nonatopic EOA had an increased frequency of exacerbations after 12-month treatment. Typically, EOA is highly associated with T2-high phenotypes, such as high eosinophilic and atopic features. However, a subset of patients with EOA exhibit nonatopic phenotypes [10]. Previous studies have reported that patients with nonatopic EOA respond poorly to inhaled corticosteroids compared with those with atopic asthma [10], suggesting that nonatopic asthma has various predisposing factors and causal mechanisms, which are possibly related to infections caused by viruses, atypical bacteria, or fungi [9,19]. Recent evidence has also demonstrated that patients with nonatopic asthma typically have more severe symptoms and require higher doses of inhaled corticosteroids for controlling symptoms [20]. In addition, our results demonstrated that in non-atopic patients, there were more EOA patients who received OCS and biologics treatment than LOA patients. The results indicate that non-atopic EOA patients may have more severe disease course and require OCS and biologics treatment than do non-atopic LOA patients. Therefore, further study is necessary to investigate the treatment response of patients with nonatopic EOA who receive standard treatment for asthma.

There is an increasing awareness of the heterogeneous features of asthma and treatment responsiveness. Given that pulmonary function decline is one of the risk factors for asthma, early identification of patients at high risk of accelerated decline in pulmonary function is important because irreversible airflow obstruction is associated with increased morbidity and mortality [21,22].

Our study has several limitations. First, the sample size was small because we included only patients who had high compliance and had regular outpatient follow-up visits for at least 1 year. The possible selection bias may occur due to recruitment of only compliant patients. Therefore, use of these results in asthma patients should be applied with caution. Second, no fractional exhaled nitric oxide (FeNO) or biomarker data were collected in our study. Third, the assessment of acute exacerbations in the previous year was based on self-reported information; thus, it could be influenced by recall bias.

It is important to distinguish between different phenotypes of asthma because treatment responses may vary. A significant proportion of patients with EOA lack the characteristics of atopy and may have an increased incidence of acute exacerbations, even under the standard treatment for asthma. Overall, identifying these phenotypes will improve prognosis and treatment guidance. Therefore, future research should focus on identifying the inflammatory pathways in nonatopic asthma and potential phenotype-guided therapies.

## Figures and Tables

**Figure 1 jcm-11-07309-f001:**
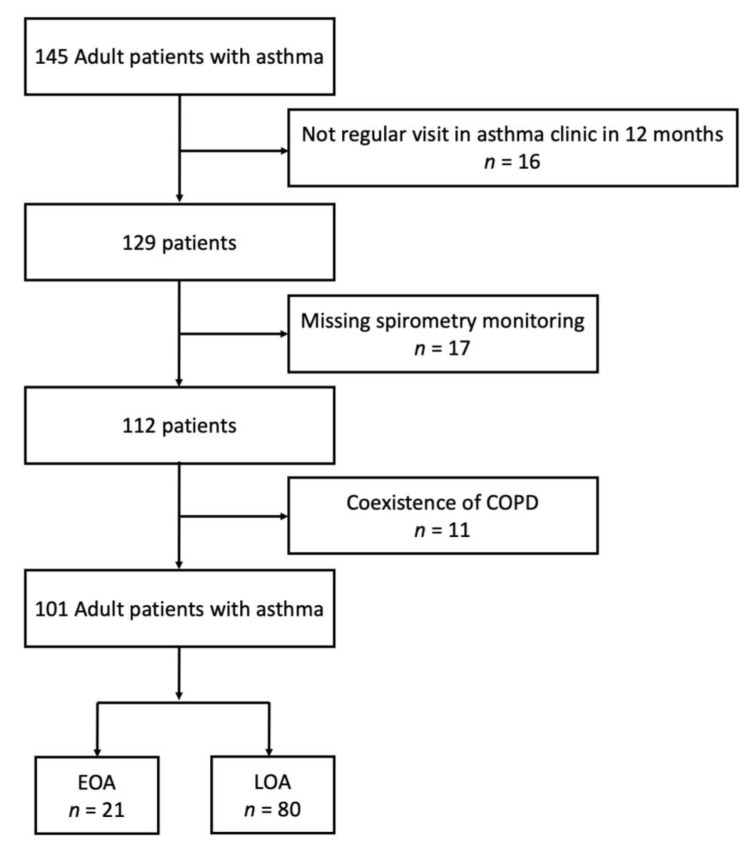
Study profile: The number of patients enrolled and analyzed in the study.

**Figure 2 jcm-11-07309-f002:**
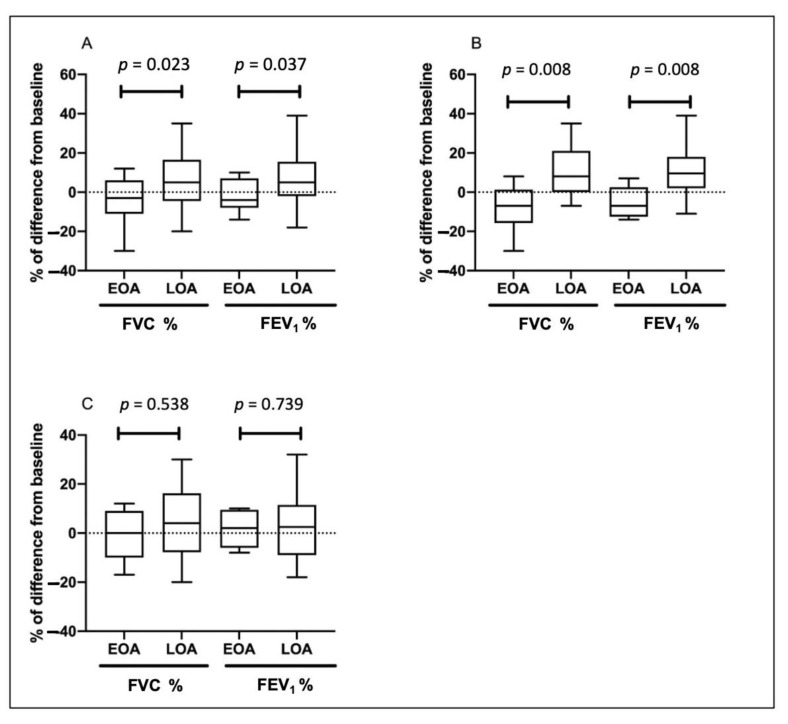
Change of forced vital capacity (FVC) and forced expiratory volume in 1 s (FEV_1_) in patients with early-onset asthma (EOA) and late-onset asthma (LOA) after 12-month treatment in the total cohort (**A**), patients <65 years old (**B**), and patients ≥65 years old (**C**). Boxplots show the median (bar), the first and third quartiles (box), and the 1st and 99th percentiles (whiskers) of the biomarkers level for each asthma status.

**Figure 3 jcm-11-07309-f003:**
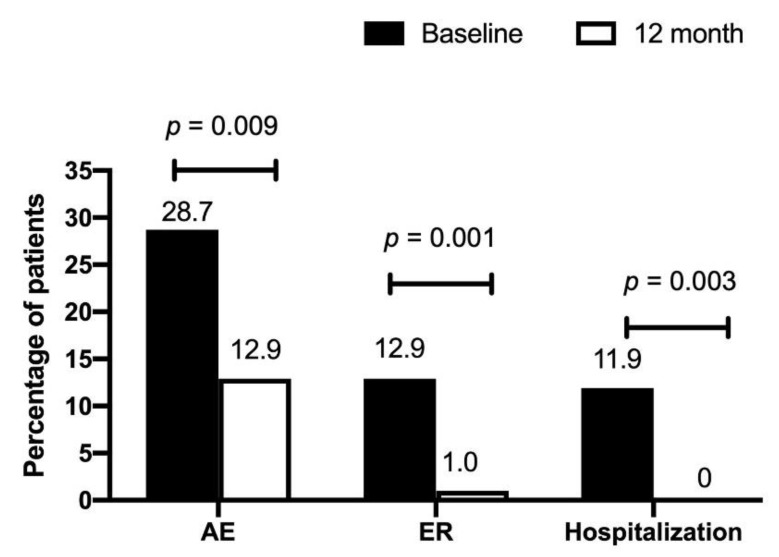
Percentage of acute exacerbation (AE), emergency room (ER) visits, and hospitalizations in patients with asthma at baseline and after 12-month treatment.

**Figure 4 jcm-11-07309-f004:**
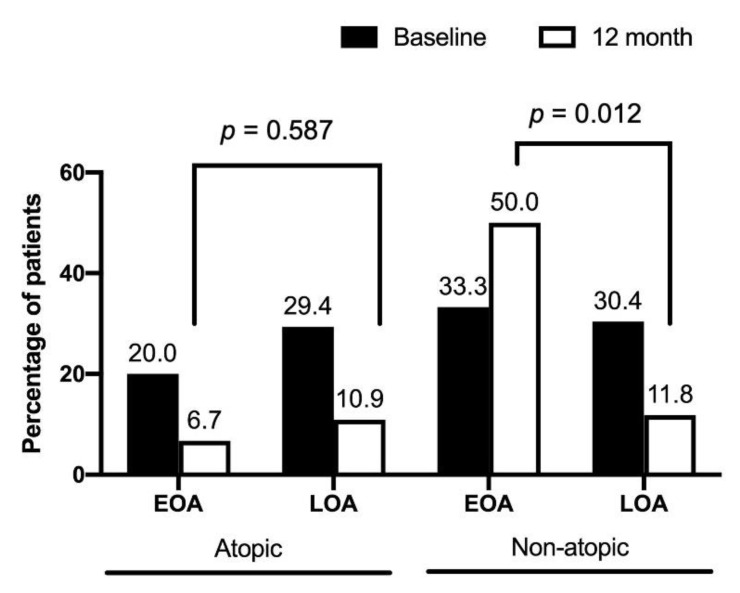
Percentage of acute exacerbations in early-onset asthma (EOA) and late-onset asthma (LOA) patients with atopic and nonatopic status after 12-month treatment. The baseline percentages of acute exacerbation were similar between EOA and LOA in patients with atopic and non-atopic status.

**Table 1 jcm-11-07309-t001:** Characteristics of patients with early-onset and late-onset asthma.

Characteristics	Total	EOA	LOA	*p*-Value	Atopic	Non-Atopic	*p*-Value
	*n* = 101	*n* = 21	*n* = 80		*n* = 49	*n* = 52	
Age, years, mean (SD)	63.7 ± 15.1	48.5 ± 18.5	67.7 ± 11.1	<0.0001	60.6 ± 16.2	66.6 ± 13.5	0.044
Male, *n* (%)	59 (58.4%)	12 (57.1%)	47 (58.8%)	0.894	23 (46.9%)	36 (69.2%)	0.023
Body mass index, kg/m^2^	25.2 ± 4.0	25.6 ± 3.2	25.2 ± 4.2	0.675	24.4 ± 4.2	26.0 ± 3.7	0.049
Non-smoker, *n* (%)	48 (47.5%)	11 (52.4%)	37 (46.3%)	0.617	25 (51.0%)	23 (44.2%)	0.495
ACT score, mean (SD)	21.3 ± 4.4	20.4 ± 5.7	21.6 ± 3.9	0.291	21.3 ± 4.8	21.3 ± 3.9	1.00
Age at asthma onset, years	53.8 ± 19.4	23.4 ± 10.8	61.8 ± 11.6	<0.0001	46.9 ± 18.6	60.4 ± 18.0	<0.0001
Duration of asthma, years	9.8 ± 14.3	25.1 ± 22.1	5.8 ± 7.5	<0.0001	13.6 ± 16.2	6.3 ± 11.2	0.009
Family history of asthma, *n* (%)	30 (29.7%)	10 (47.6%)	20 (25.0%)	0.044	13 (26.5%)	17 (32.7%)	0.498
Comorbidities							
Gastroesophageal reflux, *n* (%)	49 (48.5%)	12 (57.1%)	37 (46.3%)	0.374	23 (46.9%)	26 (50.0%)	0.758
Allergic rhinitis, *n* (%)	61 (60.4%)	18 (85.7%)	43 (53.8%)	0.008	36 (73.5%)	25 (48.1%)	0.009
Rhinosinusitis with or without polyps, *n* (%)	26 (25.7%)	10 (47.6%)	16 (20.0%)	0.010	14 (28.6%)	12 (23.1%)	0.528
Aspirin sensitivity, *n* (%)	7 (7.0%)	3 (14.3%)	4 (5.0%)	0.136	5 (10.2%)	2 (3.8%)	0.209
Anxiety or depression, *n* (%)	49 (48.5%)	8 (38.1%)	41 (51.2%)	0.283	19 (38.8%)	30 (57.7%)	0.057
Obstructive sleep apnea, *n* (%)	26 (25.7%)	3 (14.3%)	23 (28.7%)	0.177	8 (16.3%)	18 (34.6%)	0.036
Allergic status							
IgE level, kU/L, mean (SD)	307.0 ± 556.2	220.6 ± 250.1	333.8 ± 620.1	0.176	480.2 ± 703.6	104.3 ± 144.2	0.001
ECP level, μg/L, mean (SD)	12.4 ± 15.1	13.8 ± 10.5	11.9 ± 16.3	0.624	12.0 ± 11.4	12.9 ± 18.7	0.786
Eosinophil count, cells/μL	224.9 ± 228.4	283.8 ± 197.6	213.9 ± 233.4	0.334	220.1 ± 154.4	227.9 ± 256.5	0.885
Atopy, *n* (%)	49 (48.5%)	15 (71.4%)	34 (42.5%)	0.018	49 (100%)	0 (0%)	<0.0001
Fungus sensitization, *n* (%)	16 (15.8%)	3 (14.2%)	13 (16.3%)	0.685	16 (32.7%)	0 (0%)	<0.0001
Baseline pulmonary function test							
FVC, % of prediction	78.0 ± 18.8	84.8 ± 14.9	76.3 ± 19.3	0.071	82.3 ± 19.1	74.1 ± 17.8	0.029
FEV_1_, % of prediction	69.6 ± 21.6	75.9 ± 21.7	68.0 ± 21.4	0.147	71.7 ± 23.6	67.9 ± 19.5	0.359
FEV_1_/FVC (%)	71.6 ± 14.1	73.1 ± 13.8	71.2 ± 14.2	0.582	71.5 ± 16.5	71.6 ± 11.6	0.951
Treatment							
ICS, *n* (%)	3 (3.0%)	1 (1.0%)	2 (2.0%)	0.708	3 (6.1%)	0 (0%)	0.070
ICS+LABA, *n* (%)	86 (75.1%)	18 (85.7%)	68 (85%)	0.935	44 (89.8)	42 (80.8%)	0.202
ICS+LABA+LAMA, *n* (%)	16 (15.8%)	3 (14.3%)	13 (16.3%)	0.826	9 (18.5%)	7 (13.5%)	0.500
Montelukast, *n* (%)	42 (41.6%)	8 (38.1%)	34 (42.5%)	0.715	20 (40.8%)	22 (42.3%)	0.879
OCS, *n* (%)	12 (11.9%)	3 (14.3%)	9 (11.3%)	0.702	7 (14.3%)	5 (9.6%)	0.468
Biologics, *n* (%)	3 (3.0%)	1 (4.5%)	2 (2.5%)	0.587	2 (4.1%)	1 (1.9%)	0.523

EOA: early-onset asthma (age of onset < 40 year of age); LOA: late-onset asthma; ACT: asthma control test; FVC: forced vital capacity; FEV_1_: forced expiratory volume in one second; IgE: immunoglobulin E; ECP: eosinophil cationic protein; Atopy: any positive specific IgE; ICS: inhaled corticosteroids; LABA: long-acting beta 2 agonist; LAMA: long-acting muscarinic antagonist; and OCS: oral corticosteroids.

**Table 2 jcm-11-07309-t002:** Clinical outcomes of patients with asthma after 12-month treatment.

	Total	EOA	LOA	*p*-Value	Atopic	Non-Atopic	*p*-Value
Variables	*n* = 101	(*n* =21)	(*n* = 80)		(*n* = 49)	(*n* = 52)	
ACT score, mean (SD)	23.2 ± 2.8	22.2 ± 5.7	23.6 ± 2.3	0.121	23.4 ± 2.8	23.4 ± 3.2	0.977
-Mean difference from baseline	2.1 ± 4.2	1.4 ± 6.4	2.2 ± 3.8	0.615	0.8 ± 1.8	2.6 ± 5.6	0.250
FVC, % of prediction	81.6 ± 22.6	74.8 ± 24.1	83.4 ± 22.2	0.266	86.2 ± 23.2	77.0 ± 21.6	0.147
-Mean difference from baseline	3.8 ± 13.9	−4.6 ± 12.0	6.1 ± 13.6	0.023	5.6 ± 12.1	2.0 ± 15.5	0.343
FEV_1_, % of prediction	70.4 ± 24.9	62.5 ± 23.8	89.6 ± 112.7	0.436	72.2 ± 28.2	68.6 ± 21.5	0.604
-Mean difference from baseline	5.0 ± 12.7	−2.1 ± 8.4	6.8 ± 13.1	0.037	6.7 ± 11.5	3.2 ± 13.8	0.329
FEV_1_/FVC (%)	68.4 ± 13.8	67.3 ± 12.1	67.7 ± 13.1	0.778	64.9 ± 12.5	71.8 ± 14.4	0.069
-Mean difference from baseline	0.8 ± 7.5	2.8 ± 10.2	0.3 ± 6.7	0.341	0.3 ± 7.3	1.3 ± 7.8	0.621
Exacerbations in the past year	29 (28.7%)	5 (23.8%)	24 (30.0%)	0.577	13 (26.5%)	16 (30.8%)	0.638
Systemic corticosteroid bursts	13 (12.9%)	3 (14.3%)	10 (12.5%)	0.828	7 (14.3%)	6 (11.5%)	0.680
Emergency room visits	13 (12.9%)	2 (9.5%)	11 (13.8%)	0.607	4 (8.2%)	9 (17.3%)	0.170
Hospitalization	12 (11.9%)	3 (14.3%)	9 (11.3%)	0.702	3 (6.1%)	9 (17.3%)	0.083
Exacerbations during the 12 m	13 (12.9%)	4 (19.0%)	9 (11.3%)	0.342	5 (10.2%)	8 (15.4%)	0.437
Systemic corticosteroid bursts	13 (12.9%)	4 (19.0%)	9 (11.3%)	0.342	5 (10.2%)	8 (15.4%)	0.437
Emergency room visits	1 (1.0%)	0 (0%)	1 (1.3%)	0.607	1 (2.0%)	0 (0%)	0.301
Hospitalization	0 (0%)	0 (0%)	0 (0%)	NA	0 (0%)	0 (0%)	NA

EOA: early-onset asthma (age of onset < 40 year of age); LOA: late-onset asthma; ACT: asthma control test; FVC: forced vital capacity; and FEV_1_: forced expiratory volume in one second.

**Table 3 jcm-11-07309-t003:** The treatment for patients with asthma.

	Atopic, *n* = 49			Non-Atopic, *n* = 52		
	EOA	LOA	*p*-Value	EOA	LOA	*p*-Value
	*n* = 15	*n* = 34		*n* = 6	*n* = 46	
ICS, *n* (%)	1 (6.7%)	2 (5.9%)	0.916	0 (0%)	0 (0%)	NA
ICS+LABA, *n* (%)	13 (86.7%)	31 (91.2%)	0.631	5 (83.3%)	37 (80.4%)	0.865
ICS+LABA+LAMA, *n* (%)	1 (6.7%)	8 (23.5%)	0.160	1 (16.7%)	6 (13.0%)	0.807
Montelukast, *n* (%)	6 (40%)	14 (41.2%)	0.938	5 (50%)	19 (41.3%)	0.685
OCS, *n* (%)	1 (6.7%)	1 (17.6%)	0.311	2 (33.3%)	3 (6.5%)	0.036
Biologics, *n* (%)	0 (0%)	2 (5.9%)	0.338	1 (16.7%)	0 (0%)	0.005

Abbreviations: EOA: early-onset asthma (age of onset <40 year of age); LOA: late-onset asthma; ICS: inhaled corticosteroids; LABA: long-acting beta 2 agonist; LAMA: long-acting muscarinic antagonist; and OCS: oral corticosteroids.

## Data Availability

The datasets analyzed during the current study are available from the corresponding author upon reasonable request.

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
