# Peer review of "Different Characteristics and Clinical Outcomes between Early-Onset and Late-Onset Asthma: A Prospective Cohort Study"

_jcm, 2022, doi:10.3390/jcm11247309_

Round 1
Reviewer 1 Report
Bing-Chen Wu et al. report the responses to treatment in patients with early-onset and late-onset asthma. The topic is of interest and the manuscript well written. However, I have the following concerns.
The difference in age between the EOA and LOA may impact the results. Age-matched groups could sort out this doubt. Statistical analysis is performed considering only two groups at a time. As the sample size of some subgroups is small, it may be worthy to evaluate if the differences are still significant considering all the subgroups.
Minor comments:
Figure 2. Boxplots would be more appropriate than negative/positive bars. However, these data are already presented in table 2.
Figure 4. statistical results may be added.
I don’t think that not including children is a limitation of this study. The possible selection bias due to the selection of compliant patients should be mentioned.
Author Response
Bing-Chen Wu et al. report the responses to treatment in patients with early-onset and late-onset asthma. The topic is of interest and the manuscript well written. However, I have the following concerns.
The difference in age between the EOA and LOA may impact the results. Age-matched groups could sort out this doubt. Statistical analysis is performed considering only two groups at a time. As the sample size of some subgroups is small, it may be worthy to evaluate if the differences are still significant considering all the subgroups.
Reply to comment: in order to investigate the impact of age on the outcomes of EOA and LOA patients, we have stratified patients by age of 65 years old in the revised results section as “To elucidate the impact of age on clinical outcomes of patients with EOA and LOA, patients were stratified as age < and ³ 65 years old. Among patients < 65 years old (n=43), there were 15 EOA patients and 28 LOA patients. The characteristics of patients with EOA and LOA are listed in supplementary table S1. In the subgroup of patients ³ 65 years old, 6 patients were EOA and 52 patients were LOA. The baseline characteristics of patients ³ 65 years old are shown in table S3. In patients < 65 years old, EOA patients had greater FEV1 (-5.3±8.0 vs. 10.2±12.3, % of the predicted value, P = .008) and FVC (-8.0±12.7 vs. 8.4±11.9, % of the predicted value, P = .008) decline than LOA patients over a 12-month period (table S2; Figure 2B). In patients ³ 65 years old, the changes of pulmonary function at 12 months were similar in EOA and LOA groups (Table S4; Figure 2C).” in page 7, line 200 to 207.In addition, we also have added the results of different age group as figure 2B and figure 2 C in the revised manuscript.
Minor comments:
Figure 2. Boxplots would be more appropriate than negative/positive bars. However, these data are already presented in table 2.
Reply to comment: In the revised manuscript, we have changed the figure 2 as boxplots format. In addition, we also have added the results of different age group as figure 2B and figure 2 C in the revised manuscript. We also have re-written the legend of figure 2 as “Change of forced vital capacity (FVC) and forced expiratory volume in 1 second (FEV1) in patients with early-onset asthma (EOA) and late-onset asthma (LOA) after 12-month treatment in the total cohort (2A), patients < 65 years old (2B), and patients ³ 65 years old (2C). Boxplots show the median (bar), the first and third quartiles (box), the 1st and 99th percentiles (whiskers) of the biomarkers level for each asthma status.”
Figure 4. statistical results may be added.
Reply to comment: We have added the p value in the revised figure 4. In addition, we also have added a sentence as “The baseline percentages of acute exacerbation were similar between EOA and LOA in patients with atopic and non-atopic status.” in the revised legend of figure 4.
I don’t think that not including children is a limitation of this study. The possible selection bias due to the selection of compliant patients should be mentioned.
Reply to comment: In the revised discussion section, we have deleted the sentence of not including children as a limitation. We have added the sentence as “Lastly, the possible selection bias may occur due to only recruitment of compliant patients.” In page 10, line 290 to 291.
Reviewer 2 Report
1. Study population
It is not clear to me how the study sample is recruited. It is based on a national program, but ended up with a quite small sample size. Is it restricted to the patients who registered the program in the hospital of the authors’ affiliation? And is it voluntary that patients register for the national program? Does the asthma diagnosis precede the registration?
2. Study objective
The last sentence in Introduction says: “This study investigated patients with EOA and LOA to determine if atopic and nonatopic asthmatics have different outcomes.” However, as the authors also described, asthma phenotypes can be distinguished by age of onset as well as by atopy status. The objective should be more clearly described in the main text as well as in the abstract.
3. Study design and methods
Along with the previous point, it is not clear to me why the authors compared only EOA and LOA for spirometry outcomes, while further compared atopy status for the exacerbation outcomes.
Moreover, if the authors aim to investigate “different responses to treatment” as indicated by the title, it should be taken into account what treatment each group receives.
4. Results
If Table 1 refers to the information at baseline (otherwise it needs better description), numbers in Table 2 seem incompatible to Table 1. For example, when ACT score for EOA was 20.4 +/- 5.7 in Table 1 and 22.2 +/- 5.7 in Table 2, then the mean difference from baseline -0.4 +/- 6.8 seems incompatible. The same concern applies to many other entries in Table 2. And I wonder if the two rows (“FEV1/FVC (%)” and the next row) were mixed up.
5. Discussion
It is not clear to me what the authors intended to discuss in the fourth paragraph (lines 213-222). Its last half seems redundant to the previous paragraph.
I find the sentence “the analysis revealed that the treatment response is associated with adequate compliance” (lines 225-226) groundless. The authors did not assess compliance.
Author Response
- Study population
It is not clear to me how the study sample is recruited. It is based on a national program, but ended up with a quite small sample size. Is it restricted to the patients who registered the program in the hospital of the authors’ affiliation? And is it voluntary that patients register for the national program? Does the asthma diagnosis precede the registration?
Reply to comment: The pay-for-performance was implemented in qualified medical centers in Taiwan by NHIA. This study recruited patients who were diagnosed as asthma and received at least 2 clinic visits within 90 days in our hospital. Patients were voluntary for enrollment in the program and signed the patient agreement consent. To clarify this point, we added the above description in the revised materials and methods section in page 2, line 87 to 90.
- Study objective
The last sentence in Introduction says: “This study investigated patients with EOA and LOA to determine if atopic and nonatopic asthmatics have different outcomes.” However, as the authors also described, asthma phenotypes can be distinguished by age of onset as well as by atopy status. The objective should be more clearly described in the main text as well as in the abstract.
Reply to comment; We have re-written the objective of the study as “The study sought to investigate the characteristics and clinical outcomes of asthma patients with phenotypes distinguished by age on onset and atopy status.” in the revised introduction (page 2, line 69 to 70) and abstract (page 1, line 12 to 13).
- Study design and methods
# Along with the previous point, it is not clear to me why the authors compared only EOA and LOA for spirometry outcomes, while further compared atopy status for the exacerbation outcomes.
Reply to comment: We agree that adding the information about patients with atopic vs. non-atopic status will provide more comprehensive results of the study. In the revised manuscript, we have added the information of atopic and non-atopic patients in the revised table 1 and 2. We also have added a paragraph about the baseline characteristics of atopic and non-atopic patients as “The characteristics of patients with atopic and non-atopic status are shown in Table 1. Atopic patients were younger, had a higher BMI, had a longer duration of asthma, had a higher rate of male gender, family history of asthma, and had more comorbidities with allergic rhinitis or obstructive sleep apnea compared with patients with LOA. No difference was observed between the 2 groups in terms of smoking habits, exacerbations in the last year, medications, or ACT score. The pre-bronchodilator FVC was higher in atopic patients than in non-atopic patients. Regarding allergic status, the IgE levels and rate of fungus sensitization were higher in the atopic group than in the non-atopic group.” in the revised result section, page 3, line 148 to 155. In addition, the outcomes of atopic and non-atopic patients were described as “In contrast, the pulmonary function and acute exacerbation at 12 months were similar in atopic and non-atopic groups (table 2).” in page 6, line 190 to 192.
# Moreover, if the authors aim to investigate “different responses to treatment” as indicated by the title, it should be taken into account what treatment each group receives.
Reply to comment: In the revised results section, we have added the results of treatment received by each subgroup of asthma patients. We have added a new table 3 to provide the information of treatments in each subgroup. “In atopic patients, there was no difference in treatment for asthma in EOA and LOA groups. In non-atopic patients, there were more EOA patients received OCS and biologics treatment than LOA patients.” We have added the above description in the revised results section in page 8, line 226 to 231.
In addition, we also have added a paragraph about the difference of treatment in the revised discussion section as “In addition, our results demonstrated that in non-atopic patients, there were more EOA patients received OCS and biologics treatment than LOA patients. The results indicate that non-atopic EOA patients may have more severe disease course and require OCS and biologics treatment than non-atopic LOA patients.” in page 10, line 273 to 276.
- Results
If Table 1 refers to the information at baseline (otherwise it needs better description), numbers in Table 2 seem incompatible to Table 1. For example, when ACT score for EOA was 20.4 +/- 5.7 in Table 1 and 22.2 +/- 5.7 in Table 2, then the mean difference from baseline -0.4 +/- 6.8 seems incompatible. The same concern applies to many other entries in Table 2. And I wonder if the two rows (“FEV1/FVC (%)” and the next row) were mixed up.
Reply to comment: We would like to thank the reviewer for pointing out these typographic errors. In the revised table 1, we have corrected these errors and updated the correct data of ACT score and FEV1/FVC (%)
- Discussion
# It is not clear to me what the authors intended to discuss in the fourth paragraph (lines 213-222). Its last half seems redundant to the previous paragraph.
Reply to comment: We agree that the last half of the fourth paragraph seems redundant to previous paragraph. In the revised manuscript, we have deleted the last half of the fourth paragraph.
# I find the sentence “the analysis revealed that the treatment response is associated with adequate compliance” (lines 225-226) groundless. The authors did not assess compliance.
Reply to comment: In the revised materials and methods section, we have added “The definition of adequate compliance was patients with adequate medications compliance by self-report of more than 80% use of prescribed medications and regular clinical visits.” in page 2, line 85 to 87.
Round 2
Reviewer 1 Report
The authors appropriately addressed all the points.
Author Response
We would like to thank the reviewer for your very helpful inputs and the revised manuscript has been markedly improved.Reviewer 2 Report
The authors revised Table 1 and added a paragraph in the results section: “Atopic patients were younger, had a higher BMI, had a longer duration of asthma, had a higher rate of male gender, family history of asthma, and had more comorbidities with allergic rhinitis or obstructive sleep apnea compared with patients with LOA. No difference was observed between the 2 groups in terms of smoking habits, exacerbations in the last year, medications, or ACT score. The pre-bronchodilator FVC was higher in atopic patients than in non-atopic patients.” Unfortunately this paragraph contains so many errors compared to the figures in the table: atopic patients are less likely male than non-atopic patients; family history of asthma does not show statistically significant difference, only to name a few. This made me wonder if the figures in table are error-free.
Thanks to the description on the treatment in the results section, it is clearer in which context the “treatment” is considered in this study. I still find the title potentially misleading, as the current title insinuates the treatment being the independent variable. I suggest revising the title, e.g. different characteristics and clinical outcomes between …
My previous concern about the sentence in the discussion section “the analysis revealed that the treatment response is associated with adequate compliance” has not been resolved. The revised manuscript clearly states, “Lastly, the possible selection bias may occur due to only recruitment of compliant patients”, later in the discussion section. It is obvious the authors did not investigate the association between compliance and treatment response.
Author Response
The authors revised Table 1 and added a paragraph in the results section: “Atopic patients were younger, had a higher BMI, had a longer duration of asthma, had a higher rate of male gender, family history of asthma, and had more comorbidities with allergic rhinitis or obstructive sleep apnea compared with patients with LOA. No difference was observed between the 2 groups in terms of smoking habits, exacerbations in the last year, medications, or ACT score. The pre-bronchodilator FVC was higher in atopic patients than in non-atopic patients.” Unfortunately this paragraph contains so many errors compared to the figures in the table: atopic patients are less likely male than non-atopic patients; family history of asthma does not show statistically significant difference, only to name a few. This made me wonder if the figures in table are error-free.
Reply to comment: We would like to thank the reviewer for pointing out these important errors. In the revised manuscript, we have carefully rechecked the results and assured that all the data in the tables were correct. We also have re-written these sentences as “Atopic patients were younger, had a lower BMI, had a longer duration of asthma, had a lower rate of male gender, and had more comorbidities with allergic rhinitis or obstructive sleep apnea compared with non-atopic patients. No difference was observed between the 2 groups in terms of smoking habits, family history of asthma, exacerbations in the last year, medications, or ACT score.” In the revised results section, page 3, line 143 to 151.
Thanks to the description on the treatment in the results section, it is clearer in which context the “treatment” is considered in this study. I still find the title potentially misleading, as the current title insinuates the treatment being the independent variable. I suggest revising the title, e.g. different characteristics and clinical outcomes between …
Reply to comment: We agree that changing the tile as Different Characteristics and Clinical Outcomes between Early-Onset and Late-Onset Asthma: A Prospective Cohort Study will be more suitable. In the revised manuscript, we have changed the title as “Different Characteristics and Clinical Outcomes between Early-Onset and Late-Onset Asthma: A Prospective Cohort Study”.
My previous concern about the sentence in the discussion section “the analysis revealed that the treatment response is associated with adequate compliance” has not been resolved. The revised manuscript clearly states, “Lastly, the possible selection bias may occur due to only recruitment of compliant patients”, later in the discussion section. It is obvious the authors did not investigate the association between compliance and treatment response.
Reply to comment: The study recruited patients with adequate compliance and did not investigate the association between compliance and treatment response. The sentence “the analysis revealed that the treatment response is associated with adequate compliance” may cause misleading. In the revised manuscript, we have deleted the sentence “However, this study provided real-world observation data, and the analysis revealed that the treatment response is associated with adequate compliance”. We also have re-written the sentence as “The possible selection bias may occur due to only recruitment of compliant patients. Therefore, use of these results in asthma patients should be applied with caution.” in the revised discussion section, page 10, line 271 to 273.